# Techno-Economic Comparison of Stationary Storage and Battery-Electric Buses for Mitigating Solar Intermittency

**DOI:** 10.3390/s23020630

**Published:** 2023-01-05

**Authors:** Arif Ahmed, Tobias Massier

**Affiliations:** TUMCREATE Ltd., Singapore 138602, Singapore

**Keywords:** photovoltaics, electric buses, solar balancing, battery ageing, optimal operation

## Abstract

The need to reduce greenhouse gas emissions from power generation has led to more and more installation of renewable energies such as wind and solar power. However, the high intermittency of these generators poses a threat to electrical grid stability. The power output of solar photovoltaic (PV) installations, for instance, depends on the solar irradiance, and consequently on weather conditions. In order to mitigate the adverse effects of solar intermittency, storage such as batteries can be deployed. However, the cost of a stationary energy storage system (SESS) is high, particularly for large PV installations. Battery electric vehicles (BEVs) are an alternative to SESS. With increasing number of BEVs, more and more storage capacity becomes available while these vehicles are charging. In this paper, we compare stationary batteries to mobile batteries of battery electric buses (BEBs) in a public bus terminus for balancing fluctuations of solar PV installations. Public buses have been chosen due to their large batteries and because they are more easily manageable than private cars. An optimisation model has been developed considering both the bus operator’s and the PV operator’s objectives. Cycle ageing of batteries is included in the investigation. Our analysis reveals that utilising public BEBs with high battery capacity to balance solar PV fluctuations can present a positive financial case.

## 1. Introduction

Photovoltaic installations have become more efficient and less expensive over the years. According to the International Energy Agency, in 2019 the installed solar photovoltaic (PV) capacity worldwide amounted to 603 GW and the total electricity generation to 681 TWh [1]. While solar PV is a viable source of power supply in many places of the world, its intermittency poses numerous challenges to the power system. Rapidly changing weather events generate power ramps by solar PV plants that can lead to short-term variations in node voltages and line loading. To maintain safe and secure power system operation in the presence of renewable generation, grid operators may impose grid constraints such as limiting the permitted ramp rate of solar PV. As a consequence, not all of the power supply from intermittent sources can fully be integrated into the grid anymore. This problem worsens for large PV installations in a small area. To remedy this, energy storage systems such as stationary batteries can be deployed in order to absorb supply spikes or demand dips, thereby maintaining smoother grid operation.

However, storage systems are expensive and require additional space. The latter can particularly be an issue in dense urban areas. With the rapid growth of battery electric vehicles (BEVs) in recent years, a high amount of mobile storage capacity is available that could be used to store energy supplied by intermittent generators and balance grid fluctuations. The main question to be answered is whether battery electric buses (BEBs) can be used instead of stationary energy storage systems (SESSs) for balancing of fluctuations of solar PV and minimising the amount of PV that cannot be integrated due to ramp rate restrictions. Further questions arising in this context are what amount of storage capacity BEBs can provide in light of the fact that they have to fulfil their schedules, and how costs and revenue relate to each other.

Therefore, we developed a model to balance fluctuations of solar PV using SESSs or BEBs. In a case study, we compare use of SESSs and BEBs in terms of the amount of PV that can be integrated and cost/revenue. As an example, we use a bus interchange and a bus depot for public buses in Singapore, a densely populated city-state with a well-developed public bus infrastructure. Singapore plans to deploy cleaner (electric) public buses [2] and solar PV on a large scale, e.g., on rooftops, as floating islands, as it is the only viable renewable source of energy in the country.

The combination of PV and SESSs has been discussed extensively in the literature. Therefore, the literature review focuses mostly on studies using BEVs for balancing of fluctuations of solar PV. Most of these studies focus on private BEVs or distributed small PV systems, and omit short-term fluctuations of solar PV supply.

A state-of-the-art system control and energy management aspects of BEV charging stations were presented in [3]. The hierarchical control of BEV charging stations was outlined, which offered decoupled control objectives in different layers of the BEV charging micro-grid system. Important contents of charging infrastructure standards, basic power architectures, energy storage technologies were highlighted. Torreglosa et al. [4] analysed and evaluated a BEV charging station supplied by PV solar panels, batteries and grid connection. A decentralised energy management system was developed for regulating the energy flow among the PV system, the battery, and the grid in order to achieve efficient charging of BEVs. Batteries were controlled by a model predictive controller in order to keep the bus voltage at its reference value. The study focused mainly on control schemes and the impacts on the electrical grid for smaller time period. Tarroja et al. [5] presented a study on comparing vehicle-to-grid (V2G) charging BEVs to SESSs. They concluded that using BEVs with V2G can improve utilisation of renewable energies and reduce greenhouse gas (GHG) emissions, but are less capable of balancing power plants.

The possibility of introducing V2G technology in a local public transport bus service in the city of Milan in Italy was presented in [6]. The authors presented a feasibility analysis comparing the energy demand of BEBs and the energy available from buses charging in two depots, as well as a financial evaluation estimating cost and revenue associated with energy trading between the two depots. They found that more incentives to convince bus operators to invest in V2G may be required in the future. However, no consideration to distributed generation such as solar PV or SESSs were taken into account. Manzolli et al. [7] developed an optimisation model to deal with the charging schedule of a fleet of BEBs, aiming to minimise the charging cost of BEB fleets including battery ageing. The authors presented a case study using real-world data from a small electric bus fleet of eleven BEBs in a medium-size Portuguese city. They concluded that below a battery replacement cost threshold of EUR 100 per kWh it may become economically attractive for public transport operators to sell back energy to the grid for a given remuneration scheme. Energy trading between the BEB fleet and the grid could soon be economically viable considering battery replacement costs may go down in the near future. The focus of this work was not on the technical feasibility, but rather on the possibilities of trading energy with the grid regarding economic aspects. Spatiotemporal energy demand of BEBs subject to traffic and weather conditions was not taken into consideration. A tool for estimating the energy and charging demand of electrified public transit using public data available for over 150 cities/states globally was presented and demonstrated in two case studies in [8]. The tool applied heuristic vehicle scheduling to model the charging profiles and electricity demand of “battery electrified public transport” for various charging regimes across different geographical scales. A case study of New South Wales, Australia, showed that the impact of battery electrified public transport was most significant in the lowvoltage network, where adding a depot for BEBs led to an increase of the annual critical peak demand at the local zone substation of up to 17%. In [9], a method to reduce PV curtailment based on an auction mechanism was presented. Private owners of BEVs who agree to shift their charging times earn an incentive. Alam et al. [10] developed a control strategy to effectively utilise the battery capacity of plug-in electric vehicles to mitigate the impact of fluctuating solar PV supply. The proposed strategy was tested using a real distribution system data in Australia. The authors approached the problem by an analytical solution as opposed to conventional optimisation approaches. Boström et al. [11] presented a study on a nationwide energy system based on PV and BEVs with V2G capabilities only. The authors showed that in theory, Spain could be supplied by such a system with sufficient installed PV and BEVs. However, a comparison with SESSs was not included in the study. Reddy and Meikandasivam [12] presented a control strategy for private BEVs to minimise fluctuations in the load profile, voltage deviation, and cost of charging. They used a multi-objective genetic algorithm and generated Pareto fronts yielding a set of optimal power transactions. A charge scheduling system for BEBs in a bus depot supported by a stationary energy system and PV was proposed in [13]. The authors considered one bus service with 24 chargers in the bus depot. The objective was to maximise the bus depot operator’s profit while avoiding overload of the feeder the chargers were connected to. A combination of a 100-KW PV installation and a 500-kWh SESS was used to achieve the objective. Depreciation costs of PV, the SESS, and the chargers were taken into account. An optimisation method for electric bus transit centres equipped with solar PV panels and SESS to mitigate the adverse impacts of fast charging was presented in [14]. Short-term fluctuations of PV were not taken into account, although battery degradation cost were considered using a model available in the literature. Brinkel et al. [15] analysed the impact of short-term PV power fluctuations and the mitigation potential of private BEVs. They considered a small low-voltage system with private homes having smart meters and private BEVs that could be charged or discharged using V2G whenever necessary. PV data was available in a temporal resolution of one data point per 2 s. The objective was to minimise the total charging costs. The authors found that PV penetration could become an issue from 2030 onwards. In the authors’ model, the PV profile was known to the model for the whole period of simulation. The energy storage potential of private BEVs in small distribution networks was analysed in [16]. The BEVs were aggregated for this purpose. Then, a load power smoothing control strategy was developed considering the BEVs’ maximum storage potential. A fuzzy logic controller to utilise private plugin-hybrid BEVs to balance fluctuations of solar PV was presented in [17]. The controller was tested on a 33-node radial distribution feeder in a real-time simulation setup. Particularly in the occurrence of fast cloud transients, the support of BEVs was necessary in order to maintain node voltages. The study focused on the technical implementation of the controller, not the system cost aspects.

The studies presented in [4,10,15,16,17] utilised high-resolution solar PV generation data to incorporate short-term fluctuations of solar. While the limited ramp rate of batteries was considered in [5,12,15], only in [7,14], was ageing/degradation of batteries taken into account; however, that paper did not consider PV ramp rates. Moreover, the comparison of SESSs versus mobile storage capacity of BEBs for balancing the ramps of large solar PV installation on the grid has not been investigated.

In this paper, we compare SESSs to public BEBs in a bus interchange for balancing short-term fluctuations of the power supply of large solar PV installations. Public BEBs are considered because (i) owners of private BEVs or BEBs might refuse to let a grid or PV operator decide when and to what extent to charge or discharge their batteries unless they receive high incentives, and (ii), the amount of energy that can be utilised for balancing solar PV fluctuations is much larger in BEBs compared to private cars with smaller batteries.

For a realistic model and comparison, four factors are incorporated: (i) limited ramp rate of batteries, (ii) cycle ageing of batteries, (iii) reduction of charging power at high state of charge (SoC), and (iv) time BEBs require to move to and away from a charger. To the best of our knowledge, no previous study has incorporated all of these four factors. Moreover, optimisation is performed both from the PV operator’s and the bus operator’s perspective, with both aiming to maximise their own revenue. Hence, the contributions of this paper can be summarised as follows:A model for balancing fluctuations of solar PV using SESSs or BEBs considering battery ageing and bus operation information obtained from a sophisticated mobility simulatorTwo different objectives for the PV and bus operator to maximise their own revenueComparison of PV curtailment and cost/revenue for both PV and bus operator for use of SESSs and BEBs with the two different objectivesRevenue comparison of different ramp rate limitsRevenue comparison for the PV operator and recommendations for both the PV and bus operator

In Section 2, the problem description is presented. The model formulation for balancing solar PV is presented in Section 3, followed by the battery ageing model in Section 4. A case study at a bus interchange in Singapore is presented in Section 5. Section 6 concludes the paper.

## 2. Problem Description

Higher penetration of renewable energy resources in distribution grids can have adverse effects on the security, reliability, and stability of the power system. Therefore, grid operators may propose ramp requirements for solar PV installations. By limiting the permitted ramped power of PV, only a part of the generated PV energy can be fed into the grid, while the rest has to be discarded unless storage systems are available. Any energy not fed into the grid due to ramping is income lost by the PV plant owner. In this paper, this is termed as the cost of ramping from the PV plant operator’s perspective.

The energy $E_{\mathrm{PV}}\left(t\right)$ generated by a PV system at any time *t* is split into one portion that is fed into the grid ($E_{\mathrm{PV}2\mathrm{Grid}}\left(t\right)$) and another portion that has to be discarded ($E_{\mathrm{PVramped}}\left(t\right)$), i.e., energy that has been “ramped” at time *t*, as provided in Equation (Equation 1).
(1)$$\forall t\in T:E_{\mathrm{PV}}\left(t\right)=E_{\mathrm{PV}2\mathrm{Grid}}\left(t\right)+E_{\mathrm{PVramped}}\left(t\right)$$

The permitted PV ramp limits are defined in Equations (Equation 2) and (Equation 3).
(2)$$\forall t\in T\setminus \left\{t_{0}\right\}:\frac{E_{\mathrm{PV}2\mathrm{Grid}}\left(t\right)-E_{\mathrm{PV}2\mathrm{Grid}}\left(t-1\right)}{E_{\mathrm{PV}2\mathrm{Grid}}\left(t\right)}\leq R_{\mathrm{PVramp},\max}$$
(3)$$\forall t\in T\setminus \left\{t_{0}\right\}:\frac{E_{\mathrm{PV}2\mathrm{Grid}}\left(t-1\right)-E_{\mathrm{PV}2\mathrm{Grid}}\left(t\right)}{E_{\mathrm{PV}2\mathrm{Grid}}}\leq R_{\mathrm{PVramp},\max}$$

## 3. Model Formulation for Balancing Ramped Solar PV Using Storage Systems

In this section, optimisation models for the the use of storage systems are formulated in order to reduce the amount of ramped PV energy. Stationary battery storage systems are compared to “mobile” batteries of BEBs available for charging. Two optimisation problems are formulated for the case of using BEBs, one considering the PV operator’s objective and one considering the bus operator’s objective. All variables, parameters, and sets used in this section are listed in Abbreviations.

### 3.1. Stationary Battery Storage

In this subsection, a stationary battery system is to be installed at the solar PV plant. The battery system operates with the solar PV plant to feed more energy back into the grid while fulfilling the ramp rate requirement.

The goal is to maximise the amount of energy that can be fed into the grid while observing the PV ramp restrictions. Battery charge/discharge ramp rates are considered in order to protect the battery from excessive degradation. The battery ramp rate is defined as the maximum allowed charge/discharge energy from one time step to the next. Similar to PV ramp rates, battery ramp rates can be expressed as a percentage of the battery capacity. The optimisation problem is formulated as follows:(4)$$\max\;\sum_{t\in T}E_{\mathrm{PV}2\mathrm{Grid}}\left(t\right)$$

The following constraints are defined:(5)$$\forall t\in T:E_{\mathrm{PV}2\mathrm{Grid}}\left(t\right)+E_{\mathrm{PVramped}}\left(t\right)+E_{\mathrm{PV}2\mathrm{Batt}}\left(t\right)=E_{\mathrm{PV}}\left(t\right)$$
(6)$$\forall t\in T:E_{\mathrm{PV}2\mathrm{Grid}}\left(t\right)+E_{\mathrm{Batt}2\mathrm{Grid}}\left(t\right)=E_{2\mathrm{Grid}}\left(t\right)$$
(7)$$\forall t\in T:0.2\leq SOC_{\mathrm{batt}}\left(t\right)\leq 0.8$$
(8)$$\forall k\in \left\{0,1,\ldots ,364\right\}:SOC_{\mathrm{batt}}\left(k\times 1440\right)=0.8$$
(9)$$\begin{array}{cc} \forall t\in T:SOC_{\mathrm{batt}}\left(t\right) & =SOC_{\mathrm{batt}}\left(t-1\right)\\ & +\eta_{\mathrm{charging}}\;\cdot \;\frac{E_{\mathrm{PV}2\mathrm{Batt}}\left(t\right)}{Q_{\mathrm{batt}}}\\ & -\frac{1}{\eta_{\mathrm{discharging}}}\;\cdot \;\frac{E_{\mathrm{Batt}2\mathrm{Grid}}\left(t\right)}{Q_{\mathrm{batt}}} \end{array}$$
(10)$$\forall t\in T:SOC_{\mathrm{batt}}\left(t\right)-SOC_{\mathrm{batt}}\left(t-1\right)\leq SOC_{\mathrm{ramp}}$$
(11)$$\forall t\in T:SOC_{\mathrm{batt}}\left(t\right)-SOC_{\mathrm{batt}}\left(t-1\right)\geq -SOC_{\mathrm{ramp}}$$
(12)$$\forall t\in T:E_{\mathrm{PV}2\mathrm{Batt}}\left(t\right)\leq B_{\mathrm{charging}}\left(t\right)\;\cdot \;Ch/Dch_{\max}$$
(13)$$\forall t\in T:E_{\mathrm{Batt}2\mathrm{Grid}}\left(t\right)\leq B_{\mathrm{discharging}}\left(t\right)\;\cdot \;Ch/Dch_{\max}$$
(14)$$\forall t\in T:B_{\mathrm{charging}}\left(t\right)+B_{\mathrm{discharging}}\left(t\right)\leq 1$$
(15)$$\forall t\in T:E_{2\mathrm{Grid}}\left(t\right)-E_{2\mathrm{Grid}}\left(t-1\right)\leq E_{\mathrm{PVramp},\max}$$
(16)$$\forall t\in T:E_{2\mathrm{Grid}}\left(t\right)-E_{2\mathrm{Grid}}\left(t-1\right)\geq -E_{\mathrm{PVramp},\max}$$

Equation (Equation 5) defines the energy generated by the solar PV plant, which is equal to the sum of the energy fed to the grid, the generated energy that is ramped, and the energy that is used to charge the battery storage system. Equation (Equation 6) defines the energy fed to the grid, which is the sum of the energy from the solar PV plant to the grid and the energy discharged from the battery to the grid. Equation (Equation 7) bounds the battery storage SOC between 20% and 80%. Equation (Equation 8) ensures that the initial battery SOC and the battery SOC at the end of every day is 80%. Equation (Equation 9) is the battery SOC balance equation. Equations (Equation 10) and (Equation 11) define the battery charge/discharge ramp rate constraints. Equations (Equation 12)–(Equation 14) ensure that the battery storage can either charge or discharge at any given time step, but not both. Equations (Equation 15) and (Equation 16) define the solar PV ramp rate constraints.

### 3.2. Battery Electric Buses

In this subsection, the batteries of BEBs are considered, i.e., the BEBs take part by charging/discharging their batteries in order to balance the solar PV generation. Here, the problem is formulated from both the PV operator’s perspective and the bus operator’s perspective, both of them having a different objective. From the bus operator’s perspective, the objective is to maximise the bus operator’s revenue. This encourages the bus operator to discharge energy from the buses to the grid to earn revenue and to buy energy from the PV operator to charge the buses at a cheaper cost compared to buying energy from the grid. Therefore, the following objective function is formulated for the bus operator:(17)$$\max\;\sum_{t\in T}\left(C_{\mathrm{Bus}2\mathrm{Grid}}\;\cdot \;E_{\mathrm{Bus}2\mathrm{GridPVTot}}\left(t\right)\;-C_{\mathrm{PV}2\mathrm{Bus}}\;\cdot \;E_{\mathrm{PV}2\mathrm{BusTot}}\left(t\right)-C_{\mathrm{Grid}2\mathrm{Bus}}\;\cdot \;E_{\mathrm{Grid}2\mathrm{Bus}}\right)$$

In Equation (Equation 17), $C_{\mathrm{Bus}2\mathrm{Grid}}$ is the cost of energy when the bus operator sells energy to the grid, $C_{\mathrm{PV}2\mathrm{Bus}}$ is the cost of energy when the PV operator sells energy to the bus operator, and $C_{\mathrm{Grid}2\mathrm{Bus}}$ is the cost of energy when the grid operator sells energy to the bus operator.

From the PV operator’s perspective, the objective of the PV operator is solely to maximise their own revenue by selling energy to the grid and the BEBs.
(18)$$\max\;\sum_{t\in T}\left(C_{\mathrm{PV}2\mathrm{Bus}}\;\cdot \;E_{\mathrm{PV}2\mathrm{BusTot}}\left(t\right)+C_{\mathrm{PV}2\mathrm{Grid}}\;\cdot \;E_{\mathrm{PV}2\mathrm{Grid}}\left(t\right)\right)$$
In Equation (Equation 18), $C_{\mathrm{PV}2\mathrm{Grid}}$ is the cost of energy sold by the PV operator to the grid.

The following constraints are formulated:(19)$$\forall i\in B,\forall t\in T_{i}:0\leq E_{\mathrm{bus}}\left(i,t\right)\leq Q_{\mathrm{batt}}$$
(20)$$\forall i\in B,\forall t\in T_{i}:-Ch/Dch_{\max}\leq E_{\mathrm{charger}}\left(i,t\right)\leq Ch/Dch_{\max}$$
(21)$$\forall i\in B,\forall t\in T_{i}:E_{\mathrm{bus}}\left(i,t\right)=E_{\mathrm{bus}}\left(i,t-1\right)+E_{\mathrm{charger}}\left(i,t\right)$$
(22)$$\forall i\in B,\forall t\in T_{i}:E_{\mathrm{charger}}\left(i,t\right)=E_{\mathrm{PV}2\mathrm{Bus}}\left(i,t\right)\;+E_{\mathrm{Grid}2\mathrm{Bus}}\left(i,t\right)-E_{\mathrm{Bus}2\mathrm{GridPV}}\left(i,t\right)$$
(23)$$\begin{array}{c} \forall i\in B,\forall t\in T_{i}:Ch/Dch_{\max}\leq E_{\mathrm{PV}2\mathrm{Bus}}\left(i,t\right)\;+\\ E_{\mathrm{Grid}2\mathrm{Bus}}\left(i,t\right)-E_{\mathrm{Bus}2\mathrm{GridPV}}\left(i,t\right)\leq Ch/Dch_{\max} \end{array}$$
(24)$$\forall i\in B,\forall t\in T_{i}:E_{\mathrm{PV}2\mathrm{Bus}}\left(i,t\right)+E_{\mathrm{Grid}2\mathrm{Bus}}\left(i,t\right)\leq B_{\mathrm{charging}}\left(i,t\right)\;\cdot \;Ch/Dch_{\max}$$
(25)$$\forall i\in B,\forall t\in T_{i}:E_{\mathrm{Bus}2\mathrm{GridPV}}\left(i,t\right)\leq B_{\mathrm{discharging}}\left(i,t\right)\;\cdot \;Ch/Dch_{\max}$$
(26)$$\forall i\in B,\forall t\in T_{i}:B_{\mathrm{charging}}\left(i,t\right)+B_{\mathrm{discharging}}\left(i,t\right)\leq 1$$
(27)$$\forall t\in T_{i}:E_{\mathrm{PV}2\mathrm{Grid}}\left(t\right)+E_{\mathrm{PVramped}}\left(t\right)+E_{\mathrm{PV}2\mathrm{BusTot}}\left(t\right)=E_{\mathrm{PV}}\left(t\right)$$
(28)$$\forall t\in T_{i}:E_{\mathrm{PV}2\mathrm{BusTot}}\left(t\right)=\sum_{i\in B}E_{\mathrm{PV}2\mathrm{Bus}}\left(i,t\right)$$
(29)$$\forall t\in T_{i}:E_{\mathrm{PV}2\mathrm{Grid}}\left(t\right)+E_{\mathrm{Bus}2\mathrm{GridPVTot}}\left(t\right)=E_{2\mathrm{GridPV}}\left(t\right)$$
(30)$$\forall t\in T_{i}:E_{\mathrm{Bus}2\mathrm{GridPVTot}}\left(t\right)=\sum_{i\in B}E_{\mathrm{Bus}2\mathrm{GridPV}}\left(i,t\right)$$
(31)$$\forall t\in T:\sum_{i\in B}B_{\mathrm{charging}}\left(i,t\right)+B_{\mathrm{discharging}}\left(i,t\right)\leq n_{\mathrm{chargers}}$$

Equation (Equation 19) ensures that the energy of the battery pack in a bus is within its capacity. Equation (Equation 20) ensures that the energy of a charger does not exceed its charging/discharging capacity. Equations (Equation 21) and (Equation 22) define the energy balance equations of the BEBs. Equation (Equation 23) ensures that the charging/discharging of a bus is within the charger capacity limit. Equations (Equation 24)–(Equation 26) ensure that an individual bus can either charge or discharge at any given time instant. Equation (Equation 27) defines the solar PV generation balance. Equation (Equation 28) defines the total solar PV generation that is charging the BEB at any given time. Equation (Equation 29) defines the energy being fed to the grid at any time which is the sum of total discharging energy of the buses and the energy from solar PV to the grid. Equation (Equation 30) defines the total charging energy of the buses coming from the solar PV plant at any time. Equation (Equation 31) ensures that at any given moment the total number of chargers in operation (charging or discharging) does not exceed the total number of chargers.

## 4. Battery Ageing Cost Model

A battery ageing cost model evaluates the cost of degradation of a battery as it charges or discharges. Because a stationary battery or batteries of BEBs carry out the balancing of solar PV, it is important to quantify the total ageing costs of these batteries.

The performance and safety of a battery is limited by the ageing mechanism it undergoes, which is a complex process of electrochemical, structural, and mechanical degradation. To accurately characterise the ageing of a battery, numerous cycling tests are performed at different charging rates.

In our research, we consider the in-house battery ageing model developed in [18], and utilise it as a building block to develop and model the ageing cost of batteries in our studies. Consequently, our analysis evaluates the battery ageing of the battery storage system and the electric buses as they participate in the solar PV balancing problem.

The batteries used in the simulation are modelled based on the Panasonic 18650 Li(NiCoMn)O_2_ cell, which has a nominal voltage of 3.6 V and a cell rating of 2.25 Ah. The estimated energy fade of a Panasonic battery cell is presented in Figure 1. The ageing cost of a 2.25-Ah battery cell can be evaluated by approximating Figure 1 with two linear functions represented in Equation (Equation 32), where $C_{\mathrm{ageing}}$ is the battery ageing cost in SGD, $C_{\mathrm{batt}}$ is the cost of the battery in SGD, $P_{\mathrm{charge}/\mathrm{discharge}}$ is the charging/discharging power, $Q_{\mathrm{initial}}$ is the initial battery capacity in kWh,$Q_{\mathrm{end}}$ is the end of life battery capacity in kWh, and $r_{\mathrm{ageing}}$ is the ageing ratio between charging and discharging of the battery.
(32)$$C_{\mathrm{ageing}}=\left\{\begin{array}{c} \frac{0.0005\;\cdot \;P_{\mathrm{charge}/\mathrm{discharge}}+0.0023\mathrm{Wh}}{\left(Q_{\mathrm{initial}}-Q_{\mathrm{end}}\right)}\;\cdot \;C_{\mathrm{batt}}\;\cdot \;r_{\mathrm{ageing}}\;\cdot \;\frac{\frac{1}{60}h\;\cdot \;P_{\mathrm{charge}/\mathrm{discharge}}}{Q_{\mathrm{initial}}}\;\\ \mathrm{for}\;0\leq P_{\mathrm{charge}/\mathrm{discharge}}\leq 8.5185W\\ \frac{0.0032\;\cdot \;P_{\mathrm{charge}/\mathrm{discharge}}-0.0207\mathrm{Wh}}{\left(Q_{\mathrm{initial}}-Q_{\mathrm{end}}\right)}\;\cdot \;C_{\mathrm{batt}}\;\cdot \;r_{\mathrm{ageing}}\;\cdot \;\frac{\frac{1}{60}h\;\cdot \;P_{\mathrm{charge}/\mathrm{discharge}}}{Q_{\mathrm{initial}}}\;\\ \mathrm{for}\;8.5185W\leq P_{\mathrm{charge}/\mathrm{discharge}} \end{array}\right.$$

## 5. Case Study

In this section, we present a case study around a bus interchange with PV installation in Singapore. In 2018, the Energy Market Authority (EMA) of Singapore published their “Consultation Paper for Proposed Modification to the Transmission Code” [19], which proposed that solar PV systems greater than 1 MWp generation capacity should be subjected to ramp rate constraints in order to maintain a stable and secure electric power grid in Singapore. Consequently, the associated cost of ramping solar PV generation is important. However, no ramp rate limit is specified in the EMA’s consultation paper.

For the case study, both solar data for Singapore and bus operation at the selected interchange were obtained. In order to perform the cost analysis of ramping solar PV generation, solar data in a resolution of one data point per minute for an entire year (2013) were collected from a weather station of the Solar Energy Research Institute of Singapore (SERIS) (https://www.seris.nus.edu.sg/ (accessed on 22 February 2022)). Various PV ramp rates were investigated and the respective cost of ramping solar PV were analysed. A solar PV generation plant of 1.35 MW is assumed, which equates to 50% of the peak electrical load of Boon Lay bus interchange if its current bus fleet was fully electric.

Figure 2 shows PV generation, ramped PV, and PV energy fed to the grid, for a ramp rate of 10% over a period of five days in Singapore in January 2013. Ramped PV energy amounts to 3.96% of the total energy delivered by the PV system for the entire year, which can be translated into a loss of revenue for the PV operator. It is understood that for lower ramp limits, the amount of ramped PV and the associated costs will be higher.

Figure 3 shows the associated costs due to ramped PV and revenue of PV to grid for an electricity price of SGD 0.2/kWh for PV.

Figure 4 shows the PV ramped energy cost in SGD for the entire year for PV ramp rates from 1% to 30%. The cost for 1% is above SGD 70 k and drops sharply for ramp rate limits of 5% and below, then the decrease slows down.

Above a ramp rate limit of 10%, the marginal revenue increases, though at decreasing rate. A ramp rate limit of at least 64.9% allows feeding back all of the power generated from solar PV, i.e., 1667 MWh, to the grid. The resulting revenue fora ramp rate limit of 64.9% is SGD 333 k. However, such a high ramp rate is unlikely to be accommodated by the grid operator due to its adverse effects on the grid, such that a storage system is required to absorb/discharge high ramp rates.

We analysed, quantified, and compared the results considering the following three aspects: (i) the amount of energy from PV utilised for charging and supplying the grid; (ii) the amount of energy from PV that cannot be utilised due to grid ramp restrictions; and (iii) the operating costs of the whole system. In our case study, the electricity price is SGD 0.16/kWh for $C_{\mathrm{Grid}2\mathrm{Bus}}$. The cost of energy sold by the PV operator to the grid $C_{\mathrm{PV}2\mathrm{Grid}}$ is set to SGD 0.2/kWh. When using BEBs, $C_{\mathrm{PV}2\mathrm{Bus}}$ is SGD 0.10/kWh and $C_{\mathrm{Bus}2\mathrm{Grid}}$ is SGD 0.20/kWh.

To calculate the battery aging costs, we modeled a battery storage system with a combination of series and parallel Panasonic 18650 cells for both the stationary batteries and the bus batteries. The capacity of the stationary battery system is varied between 13.5 kWh and 67.5 kWh operating at 220 V, while the battery packs in the BEBs have a capacity of 250 kWh operating at 800 V. As discussed in Section 3.1, the batteries’ rate of change of SoC per time step is limited to alleviate battery degradation.

In the first part of the case study, a stationary battery system is used to store PV power that could not be integrated due to ramp rate limitations. In the second part, BEBs are used with the two objectives presented in Equations (Equation 17) and (Equation 18). Finally, we present a revenue comparison for the PV operator. All numbers in tables are rounded to a maximum of three figures.

### 5.1. Use of Stationary Battery Storage

Five values of battery capacity are investigated, from 1% capacity (13.5 kWh) of the solar PV plant up to 5% capacity (67.5 kWh) in steps of 1%.

The battery SoC, charging SoC, and discharging SoC for a 13.5-kWh battery with a battery ramp limit of 10% installed at the solar PV plant at the bus interchange over a period of five days is presented in Figure 5.

Figure 6 presents the PV ramp cost for the entire year for the five chosen battery capacities and maximum permitted PV ramp rates of 1% to 15%. As expected, as the battery capacity and the permitted ramp rate increase, the PV ramp cost decreases as more energy can be fed directly into the grid. The highest reduction in PV ramp cost is observed for small battery capacities and lower PV ramp rates.

In Figure 7, the revenue generated from energy fed to the grid for the entire year is presented without storage system and for the five chosen battery capacities for maximum permitted PV ramp rates of 1% to 16%. As expected, as the battery capacity increases and as the PV ramp rate increases, more energy is fed to the grid, generating increasing revenue. As the PV ramp rate increases, the total energy fed into the grid saturates and the marginal revenue decreases. The highest increase in revenue from energy to grid is observed at lower battery capacities and lower PV ramp rates.

From Figure 6 and Figure 7, it makes more sense for a PV plant owner to invest in a battery storage system when the grid code employs a very strict PV ramp rate requirement. This is due to the fact that the decrease in ramp cost and the increase in revenue from energy feed to the grid potentially offsets the battery investment cost at lower PV ramp requirements.

In Figure 8, the PV ramp cost for the entire year is presented for a 13.5-kWh battery for various battery ramp limits over the PV ramp rate limit. When the battery ramp limit is increased from 10% to 20%, a significant drop in PV ramp cost is achieved. Beyond 40%, however, no significant benefit is achieved; rather, it could be detrimental to the battery due to degradation. A similar depiction is observed in Figure 9, with increasing energy-to-grid revenue for increasing battery ramp limit. Beyond a certain battery ramp limit, the marginal increase in revenue may not offset the cost of battery degradation.

### 5.2. Use of BEBs

We chose the Boon Lay bus interchange in the West of the city-state of Singapore for this study because it is one of the biggest bus interchanges in Singapore in terms of number of bus services and available area for PV installations. Information on public bus operations is available from the Land Transport Authority (LTA) (https://www.lta.gov.sg/ (accessed on 7 February 2022)). For a realistic picture of bus operations over one day, we used the tool CityMoS (https://www.citymos.net (accessed on 25 May 2022)) [20], a microscopic large-scale agent-based mobility simulator developed at TUMCREATE. Using this tool together with information on bus operation from 2021 allowed us to acquire a realistic picture of bus operation at Boon Lay interchange considering both delays in operation, e.g., during the rush hour, and different weather conditions during the day. Both delays and hot weather lead to increased energy demands on the buses due to the air-conditioning system, which is always running in the hot and humid climate of Singapore. Charging of BEBs is implemented in a realistic way such that the maximum charging power decreases when the SoC of a bus battery reaches 80% during charging. Moreover, the time it takes for a BEB to move to a charger and away from it is considered. Hence, CityMoS provides a much more realistic estimate of the amount of energy required for every trip than synthetic driving profiles based on historic data.

Bus data for seven days were collected. Buses arrive at the interchange, charge their batteries, and start their next trip according to the schedules generated from CityMoS. While buses are present at the interchange and connected to a charger, they can balance solar PV generation by taking energy from the PV plant or by feeding energy back to the grid. Every bus is able to start charging two minutes after arrival at the interchange, and has to stop charging two minutes before departure from the interchange. The dataset contains a total of 261 unique buses generating more than 3000 trips during the seven days.

Figure 10 and Figure 11 present the total energy provided by the PV system, the amount of ramped PV energy, and the total energy fed to the grid at a 10% PV ramp limit for the bus operator’s objective (Equation (Equation 17)) and the PV operator’s objective (Equation (Equation 18)).

Table 1 shows the amount of ramped PV, PV and grid energy fed into the BEBs’ batteries, PV energy fed into the grid, energy from the BEBs’ batteries fed into the grid, and the total amount of energy fed into the grid, while Table 2 shows the incurred cost in SGD.

The stricter the ramp rate limit, the higher the amount of PV ramped, and consequently the higher the ramping cost. Using the PV operator’s objective, as little PV energy as possible is ramped; however, when using the bus operator’s objective the amount of ramped PV is rather high. This is because the bus operator tries to sell as much energy as possible to the grid, as the revenue for selling energy to the grid is higher than the cost. Because the ramp limit applies not only to PV, but to all energy fed into the grid, a significant amount of PV cannot be fed into the grid, as the BEBs already feeds in quite a lot. Using the PV operator’s objective, most PV energy is directly fed into the grid, as the grid offers a higher electricity price than the bus operator. Only PV energy that would have to be discarded is fed into the BEBs.

Consequently, the cost of ramped PV is higher when using the bus operator’s objective. For a strict ramp rate limit of 1% it is four times as high, while for a ramp rate limit of 10% it is more than 50 times as high. On the other hand, the bus operator can sell much more energy to the grid while having to buy less energy from the grid. Therefore, it is important to look at electricity cost for the bus operator and the revenue for the PV operator. Table 3 shows the cost for the bus operator and the revenue (as negative cost) for the PV operator for both objectives and different ramp rate limits.

Obviously, costs are lower and revenue is higher for higher ramp rate limits, as more PV energy can be utilised. The bus operator’s costs are about 10% lower when using the bus operator’s objective, while the PV opeator’s revenue is between 50% and 70% higher when using the PV operator’s objective.

### 5.3. Revenue Comparison for the PV Operator

Table 4, Table 5 and Table 6 show the PV operator’s revenue for all considered cases—no battery system, stationary battery with a capacity of 13.5 kWh, BEBs—and a ramp rate limit of 10%, 5%, and 1% respectively, considering both objectives. Table 7 shows the same for a ramp rate limit of 1% and a battery capacity of 27 kWh.

For ramp rate limits of 5% or 10%, the higher revenue due to a higher amount of PV that can be integrated does not outweigh the cost of the stationary battery. For a ramp rate limit of 1%, the total revenue increases with a stationary battery (Table 6). Adding a second battery (27 kWh) leads to another increase in revenue, namely, about SGD 200 (Table 7). It should be noted that the stationary battery with the PV plant charges only with the electricity from the PV generation. Subsequently, when the PV plant feeds energy into the grid it combines both generation and stored energy from the stationary battery. Therefore, in order to avoid double counting, we have avoided PV to battery cost in Table 4, Table 5, Table 6 and Table 7.

With BEBs, the revenue depends heavily on the objective. The PV operator can significantly increase its revenue when using BEBs and the PV operator’s objective. However, when using the bus operator’s objective, the PV operator’s revenue is lower as compared to using a stationary battery, or, except for a ramp rate limit of 1%, no battery at all. Hence, the bus operator and PV operator have to agree on a compromise between their objectives. For example, a potential compromise could be that the PV operator rebates part of the excessive profit to the bus operator and thus avoids having to purchase and operate a battery storage system. Similarly, the bus operator could rebate from its additional savings by avoiding purchase of electrical energy from the grid at higher cost.

## 6. Conclusions

In this manuscript, we have compared stationary batteries to mobile batteries of BEBs in a public bus interchange for mitigating intermittency of solar PV and reducing PV curtailment. An optimisation problem was developed for these purposes. Cycle battery ageing was taken into account. Bus arrival and departure times, including the SoC of their batteries, was obtained from simulation software that is able to incorporate different traffic conditions at different times of the day. For the use of BEBs, we formulated two different objective functions for the PV and bus operator.

When using SESSs, a battery system of 2% of the installed PV capacity halves PV ramp costs compared to not using a storage system at all. For a pv ramp rate limit of above 2% and a battery ramp rate limit of above 30%, the PV operator’s revenue increases only at a small pace.

When using BEBs, the amount of PV ramped depends mainly on the objective. The bus operator’s objective leads to a higher amount of ramped PV, as it is more profitable to sell energy from the bus batteries to the grid which limits the amount of PV power that can be fed in. For the PV operator, however, this model is less profitable than using no storage system at all, whereas using the PV operator’s objective increases the PV operator’s revenue by 50% to 70%. On the other hand, this reduces the bus operator’s revenue by approximately 10%. Therefore, the PV operator and bus operator should find a compromise between their objectives such that both benefit equitably.

While our investigation presents a realistic case, there are limitations. An example are unforeseeable events such as disruption of the bus service, e.g., accidents. However, CityMoS incorporates many factors such as delays during the rush hour, making the result quite close to real-life bus operations under normal conditions. We did not consider different battery capacities of the BEBs, because the decision on which BEB models to use in the future has not yet been made in Singapore. Similarly, extreme weather events resulting in high fluctuations and uncertainty in generation were not considered in this study. In general, addressing the various aspects of uncertainties can be expected to improve the results and is the recommended direction for future work.

## Figures and Tables

**Figure 1 sensors-23-00630-f001:**
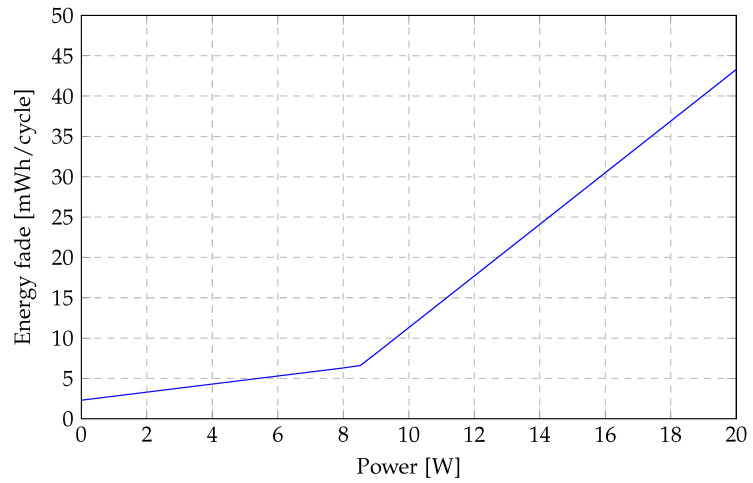
Battery energy fade in kWh/cycle.

**Figure 2 sensors-23-00630-f002:**
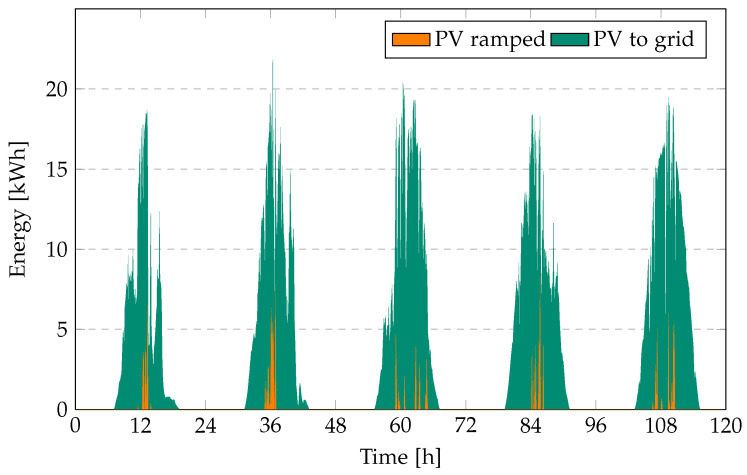
PV generation, PV ramped energy, and energy fed to the grid at 10% ramp requirement over a period of five days.

**Figure 3 sensors-23-00630-f003:**
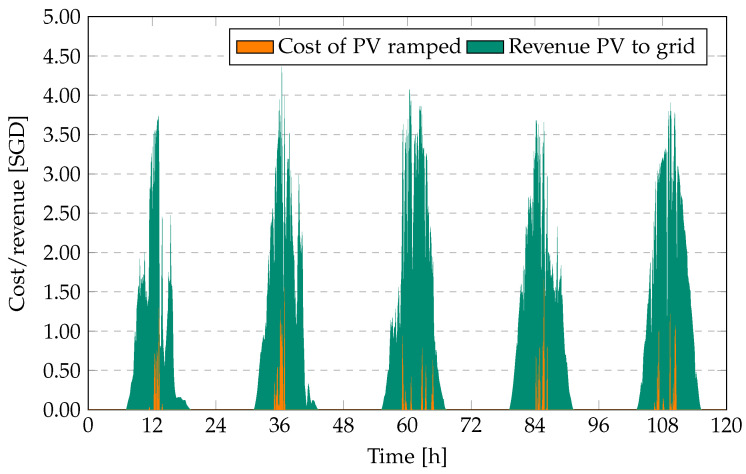
Revenue PV to grid (SGD) and cost of PV ramped (SGD) at 10% ramp requirement over a period of five days.

**Figure 4 sensors-23-00630-f004:**
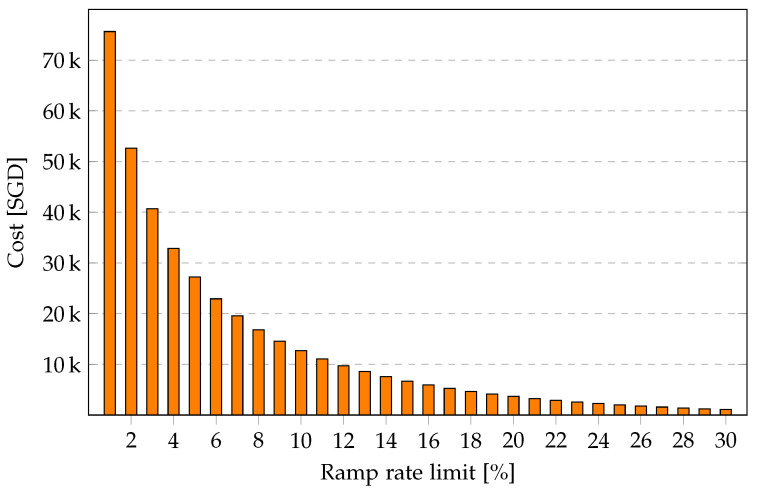
PV ramped cost (SGD) over PV ramp rate limit for one year.

**Figure 5 sensors-23-00630-f005:**
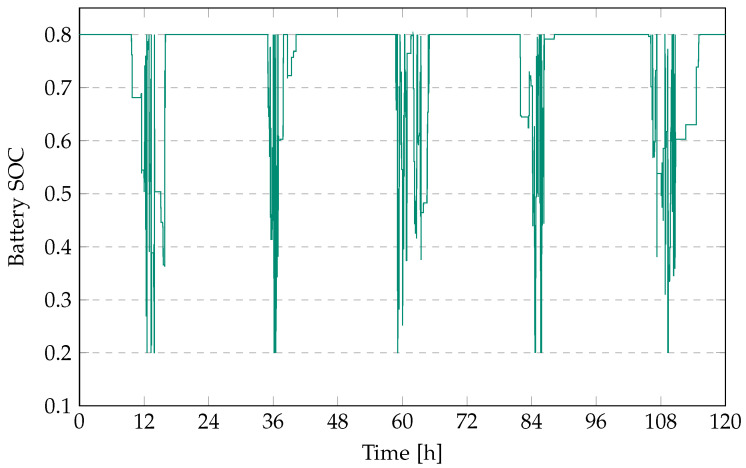
The battery SoC for a 13.5-kWh battery over a period of five days.

**Figure 6 sensors-23-00630-f006:**
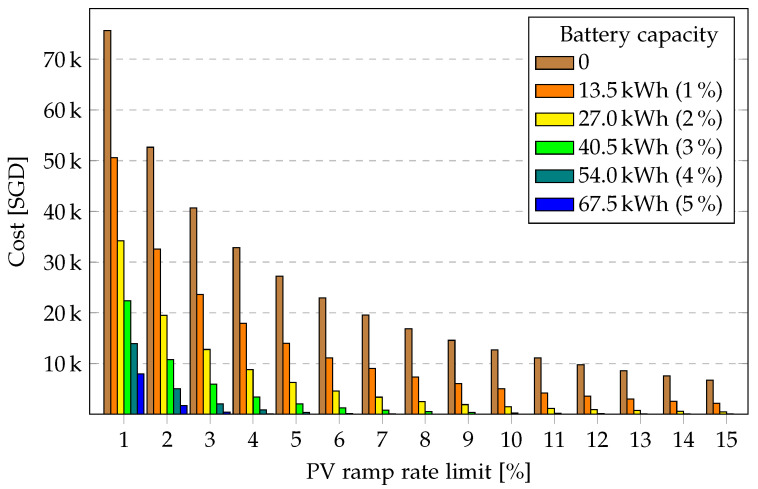
PV ramp cost over PV ramp rate limit for various battery capacities for one year.

**Figure 7 sensors-23-00630-f007:**
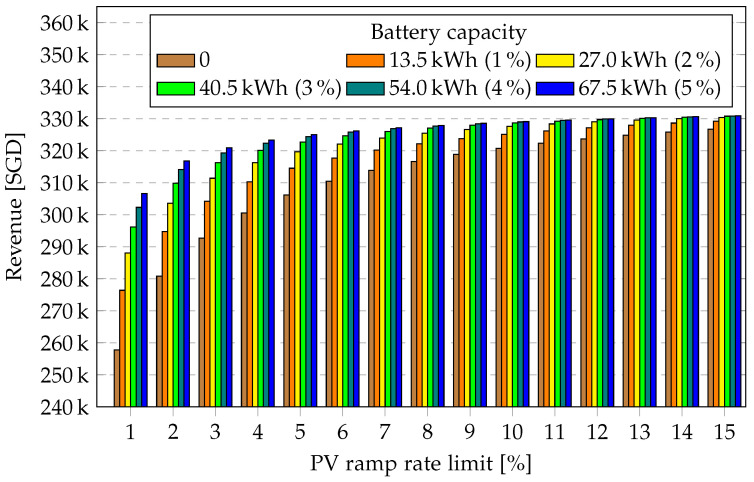
Revenue from PV to grid over PV ramp rate limit for various battery capacities for one year.

**Figure 8 sensors-23-00630-f008:**
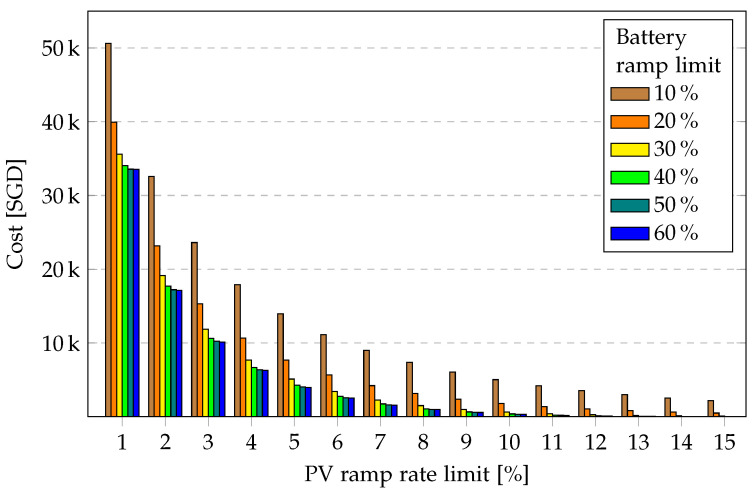
PV ramp cost over PV ramp rate limit for various battery ramp limits for one year.

**Figure 9 sensors-23-00630-f009:**
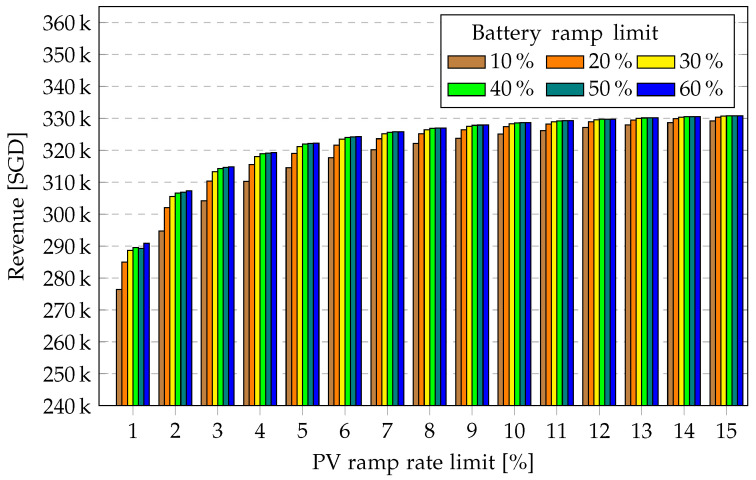
Revenue from PV to grid over ramp rates for various battery ramp limits for one year.

**Figure 10 sensors-23-00630-f010:**
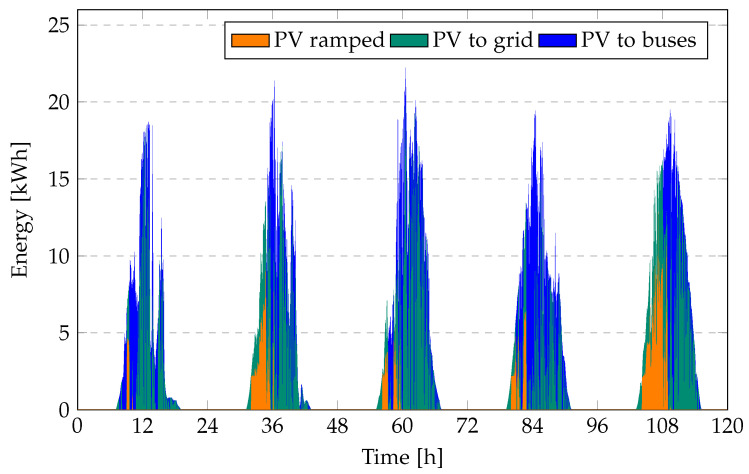
PV generation, PV 2 Bus, ramped energy, and energy fed to the grid at 10% ramp rate limit for the bus operator’s objective (Equation (Equation 17)) over a period of five days.

**Figure 11 sensors-23-00630-f011:**
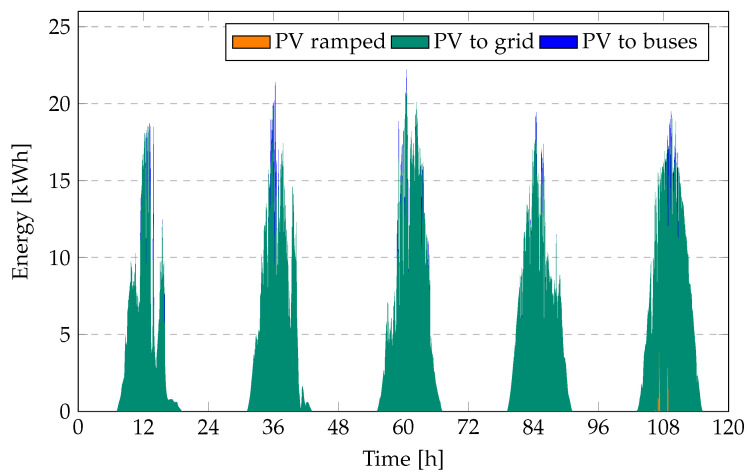
PV generation, ramped power, and energy fed to the grid at 10% ramp rate limit for the PV operator’s objective (Equation (Equation 18)) over a period of five days.

**Table 1 sensors-23-00630-t001:** Amount of PV ramped, PV energy fed to buses, grid energy fed to buses, PV energy fed to grid, energy from BEB batteries fed to grid, and total energy fed to the grid (all in kWh) for both objectives and three different ramp rate limits for one week.

	Bus Operator			PV Operator		
**Ramp Limit**	**10%**	**5%**	**1%**	**10%**	**5%**	**1%**
PV ramped	3290	3300	3570	62.0	180	874
PV to bus	17,400	17,400	18,700	803	1920	6530
Grid to bus	184,000	184,000	183,000	201,000	200,000	197,000
PV to grid	16,900	16,900	15,300	36,700	35,500	30,200
Bus to grid	17,100	13,500	9380	3610	1820	340
Total to grid	34,000	30,400	24,700	40,300	37,300	30,500

**Table 2 sensors-23-00630-t002:** Cost of PV ramped, PV energy fed into buses, grid energy fed into buses, PV energy fed into the grid, energy from BEB batteries fed into the grid, total energy fed into the grid, and battery ageing (all in SGD) for both objectives and three different ramp rate limits for one week.

	Bus Operator			PV Operator		
**Ramp Limit**	**10%**	**5%**	**1%**	**10%**	**5%**	**1%**
PV ramped	657	661	713	12.4	36.0	175
PV to bus	1740	1740	1870	80.0	192	653
Grid to bus	29,500	29,500	29,300	32,200	32,000	31,600
PV to grid	3380	3380	3060	7340	7090	6030
Bus to grid	3430	2700	1880	723	363	68.0
Total to grid	6810	6080	4930	8060	7450	6100
Battery ageing	37.0	36.0	36.0	35.0	34.0	34.0

**Table 3 sensors-23-00630-t003:** Total cost incurred for bus operator and total revenue for the PV operator (as negative cost) for both objectives for one week.

	Bus Operator			PV Operator		
**Ramp Limit**	**10%**	**5%**	**1%**	**10%**	**5%**	**1%**
Bus op.’s cost	27,800	28,500	29,200	31,500	31,900	32,200
PV op.’s cost	−4460	−4450	−4210	−7410	−7250	−6510

**Table 4 sensors-23-00630-t004:** Revenue comparison for the PV operator considering no storage, stationary batteries (1% of the installed PV capacity i.e., 13.5 kWh), and BEBs (in SGD) for a ramp rate limit of 10% for one week.

	No Battery	Stat. Battery	BEBs	
			**Bus op.**	**PV op.**
Stationary battery	–	−65.0	–	–
Battery ageing	–	−60.0	−37.0	−35.0
PV to bus	–	–	1740	80.0
PV to grid	6170	6250	3380	7340
Total revenue	6170	6130	5080	7380

**Table 5 sensors-23-00630-t005:** Revenue comparison for the PV operator considering no storage, stationary batteries (1% of the installed PV capacity i.e., 13.5 kWh), and BEBs (in SGD) for a ramp rate limit of 5% for one week.

	No Battery	Stat. Battery	BEBs	
			**Bus op.**	**PV op.**
Stationary battery	–	−130	–	–
Battery ageing	–	−101	−36.0	−34.0
PV to bus	–	–	1740	192
PV to grid	5890	6050	3380	7090
Total revenue	5890	5820	5080	7250

**Table 6 sensors-23-00630-t006:** Revenue comparison for the PV operator considering no storage, stationary batteries (1% of the installed PV capacity i.e., 13.5 kWh), and BEBs (in SGD) for a ramp rate limit of 1%. The ramping costs are not deducted from the total revenue, as they are implicitly included in the lower revenue for PV to battery, bus, or grid for one week.

	No Battery	Stationary Battery	Battery-Electric Buses	
			**Bus op.**	**PV op.**
Stationary battery	–	−130	–	–
Battery ageing	–	−120	−36.0	−34.0
PV to bus	–	–	1870	653
PV to grid	4960	5320	3060	6030
Total revenue	4960	5070	4890	6650

**Table 7 sensors-23-00630-t007:** Revenue comparison for the PV operator considering no storage, stationary batteries (2% of the installed PV capacity i.e., 27 kWh), and BEBs (in SGD) for a ramp rate limit of 1%. The ramping costs are not deducted from the total revenue, as they are implicitly included in the lower revenue for PV to battery, bus, or grid for one week.

	No Battery	Stationary Battery	Battery-Electric Buses	
			**Bus op.**	**PV op.**
Stationary battery	–	−130	–	–
Battery ageing	–	−93.0	−36.0	−34.0
PV to bus	–	–	1870	653
PV to grid	4960	5540	3060	6030
Total revenue	4960	5320	4890	6650

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
