# Peer review of "Techno-Economic Comparison of Stationary Storage and Battery-Electric Buses for Mitigating Solar Intermittency"

_sensors, 2023, doi:10.3390/s23020630_

Round 1
Reviewer 1 Report
This paper investigated the “Techno-Economic Comparison of Stationary Storage and Battery-Electric Buses for Mitigating Solar Intermittency”. However, it is not suited to this journal. I do not think that the present manuscript is worth publishing in Sensors journal.
Comments:
1- Abstract is not appropriate.
2- The literature review should be improved, and the author should investigate more research works in this area and add to the introduction.
3- The paper has not any information about the model description and model dimensions.
4- The results are not sufficient.
5- Re-write and reframe the sentences throughout the manuscript to improve the quality of the paper.
Author Response
We would like to thank the editors and reviewers for providing their review of our paper. In the following, please find our responses. The reviewers’ comments are given in blue, our responses are given in black.

Reviewer 2 Report
Good and very timely paper addressing one of the most important issues of battery’s financial optimisation.
Author Response

(The authors gave the same response as above.)

Reviewer 3 Report
This paper compared stationary batteries to mobile batteries of battery electric buses in a public bus terminus for balancing fluctuations of solar PV installations. To balance solar PV, public buses were chosen because they have more available energy in their batteries and they are more accessible and manageable than private cars. An optimization model was also developed, due to the goal of both the bus operator’s and the PV operator’s objectives. Finally, your analysis indicates that balancing the fluctuations of solar PV with large mobile storage can lead to better financial results.
However, there are some problems, which must be solved before it is considered for publication.
â… . Some sentences contain grammatical mistakes or are not complete sentences, such as, in page 1, Abstract, “easily” would be “more easily”, and in line 56, “pareto” could be “Pareto”.
â…¡. More detailed research objectives of this paper need to be supplemented in Introduction.
â…¢. Some of the parameters and variables covered in this paper are too long in the corner, and it would be better if you could simplify them.
â…£. The layout of the text and figures in this paper can be optimized.
â…¤. Conclusion needs more in it, as it’s more of an afterthought. You are suggested to highlight important findings and include afterthoughts of this work.
â…¥. The contributions of this paper are not expounded sufficiently. You need to highlight this paper’s innovative contributions.
â…¦. The paper [Hierarchical Operation of Electric Vehicle Charging Station in Smart Grid Integration Applications - An Overview] may be helpful to you. You can cite it to better your paper.
Author Response

(The authors gave the same response as above.)

Round 2
Reviewer 1 Report
The paper has not any information about the model description and model dimensions.
The results are not sufficient.
Author Response
Thank you for reviewing our paper again.
1) In the first round of revision, we elaborated on the model description. In our view, it is complete.
For further additions, we kindly ask the reviewer to provide us with more details on what is missing in which part of the model (battery ageing model, solar balancing model, bus operation model, etc.).
We added a reference to a paper that describes CityMoS on page 12, Section 5.2, first paragraph, line 317.
2) Similarly, we would require more details on why the results are not sufficient and what should be added.
We presented results on the amount and cost of PV ramped without storage system, with stationary storage system, with battery electric buses, as well as revenue for PV and bus operator for different ramp rate limits. A revenue comparison for the PV operator was included.
Reviewer 3 Report
All the issues have been solved.
Author Response
Thank you very much again for your review and your positive feedback.